# Effect of Different Decontamination Methods on Fracture Resistance, Microstructure, and Surface Roughness of Zirconia Restorations—In Vitro Study

**DOI:** 10.3390/ma16062356

**Published:** 2023-03-15

**Authors:** Rama A. Darwich, Manal Awad, Ensanya A. Abou Neel

**Affiliations:** 1Preventive and Restorative Dentistry Department, College of Dental Medicine, University of Sharjah, Sharjah P.O. Box 27272, United Arab Emirates; 2UCL Eastman Dental Institute, Biomaterials & Tissue Engineering, Royal Free Hospital, Rowland and Hill Street, London NW3 2QG, UK

**Keywords:** zirconia, decontamination, fracture resistance, microstructure, roughness

## Abstract

This study aimed to evaluate the effect of seven different decontamination methods (water, ZirClean^TM^, 37% phosphoric acid, 9.5% hydrofluoric acid, Al_2_O_3_ sandblasting, low-speed dental stone, and high-speed dental stone) on the fracture resistance, microstructure, and surface roughness of monolithic and multilayered zirconia. The as-received and sandblasted zirconia was used as a control. One-way ANOVA and *t*-test were performed. As-received monolithic zirconia was stronger (856 ± 94 MPa) than multilayered zirconia (348.4 ± 63 MPa). Only phosphoric acid (865 ± 141 MPa) and low-speed dental stone (959 ± 116 MPa) significantly increased the flexural strength of sandblasted monolithic zirconia (854 ± 99 MPa), but all tested decontamination methods except phosphoric acid (307 ± 57 MPa) and Al_2_O_3_ (322 ± 69 MPa) significantly increased the flexural strength of sandblasted multilayered zirconia (325 ± 74 MPa). Different decontamination methods did not significantly affect the flexural modulus, but introduced irregularities in the crystal as well as deep surface flaws in both types of zirconia. The surface of sandblasted monolithic zirconia is more resistant to change than multilayered zirconia. Among different decontamination methods, a low-speed dental stone could be beneficial as it significantly increased the surface roughness and fracture resistance of both types of zirconia.

## 1. Introduction

Due to its superior mechanical properties, zirconia has become an excellent alternative to metal–ceramic restorations, but due to its high opacity it was used as a framework that is typically veneered with a glass–ceramic material [1]. It has been shown that these bilayered restorations are highly susceptible to chipping, delamination, or veneer fracture [2]. To overcome these drawbacks, monolithic zirconia was introduced [3], and to improve its translucency changing the yttria content and sintering temperature was attempted [4]. As observed, an increase in yttria content increased the translucency but reduced the strength. Therefore, a new monolithic multilayered zirconia with different colors and translucency gradients was developed. This type of zirconia consists of an outer enamel, two transition layers, and an inner body layer. Across these layers, the degree of translucency decreases from the enamel to the body layer while the color intensity increases [5,6].

Zirconia exists in three crystalline forms which are monoclinic, tetragonal, and cubic [7,8]. Zirconia is non-etchable due to its pure crystalline structure [9]. In addition to its low surface energy and poor wettability, bonding is also an issue [10]. Accordingly, conditioning the fitting surface of zirconia restorations is required to achieve strong and durable restorations by increasing the surface area available for bonding [11]. During try-in, zirconia restorations can get contaminated with blood, saliva, and other contaminants [12]. This condition might adversely affect its adhesion to the tooth structure. Therefore, after try-in, the fitting surface of zirconia is usually decontaminated with the appropriate agent indicated for each material. 

In the literature, there is no clear decontamination method before the final cementation of zirconia restorations [13]. In a chair-side setting, decontamination of zirconia restoration is usually carried out using water, acetone, 70% ethanol, 5% sodium hypochlorite, 2% chlorhexidine, 96% isopropanol, cleaning agent [14], phosphoric acid, ethyl cellulose-based paints, or plasma treatment [12]. There is a misconception that the previous methods would not be effective enough in cleaning zirconia surfaces. So, there is always a tendency to use air abrasion, high- and low-speed dental stones, or even hydrofluoric acid to ensure proper decontamination and hence good bonding. These methods, however, might affect the crystalline structure of zirconia and introduce more roughness that will adversely affect its fracture resistance.

### Aim of the Study

This study aimed to investigate the effect of different decontamination methods including water, ZirCleanTM (BISCO, Inc. Schaumburg, IL, USA), phosphoric acid (DENU ETCH-37 by HDI Inc., Seoul, Republic of Korea), hydrofluoric acid (BISCO, Inc. Schaumburg, IL, USA), sandblasting (Al_2_O_3_, Zhermack, Polesine (RO), Italy), low-speed dental stone (Dentsply Sirona, Charlotte, NC, USA), and high-speed dental stone (Dentsply Sirona, Charlotte, NC, USA) on the fracture resistance, microstructure, and surface roughness of sandblasted monolithic and multilayered zirconia. 

## 2. Materials and Methods

Monolithic (Kyocera Fineceramics Precision GmbH from StarCeram Z, Lorenz-Hutschenreuther, Germany) and multilayered zirconia disks (Kyocera Fineceramics Precision GmbH from StarCeram Z, Lorenz-Hutschenreuther, Germany) were used. Their shrinkage factors (SF) were 1.266 and 1.233, respectively. 

### 2.1. Sample Size Calculation

Based on previous research [15,16], sample size was determined to be 10 samples per group (*n* = 10). Since 9 groups from each material were tested, the total number of samples was therefore 180 (90 samples from each material). 

### 2.2. Sample Preparation

A total of 180 rectangular (25 × 5 × 2.5 mm^3^) samples were used (*n* = 90 per material; *n* = 10 per group). Samples were prepared from zirconia disks using imes-icore CORiTEC 150i pro milling machine (Dental depot, GmbH, Rudolf-Diesel-Straße-8, 85221, Dachau, Germany) after considering the SF of each material. Samples were milled and sintered at 1400 °C for 3 min using a furnace (mv-r, MIHM-VOGT, Friedrich-List-Straße, Germany) to achieve their maximum strength. Ten samples from each material were randomly selected to be tested immediately as the as-received group (G0). The rest of samples were air-blasted on one surface with 50 µm Al_2_O_3_ at 2.5 bars for 15 s, 10 cm distance, and perpendicular to the surface. Then, another 10 air-blasted samples from each material were randomly selected to be tested immediately as sandblasted/uncontaminated group (G1). Both G0 and G1 were used as controls. The rest of the samples were kept in saliva, obtained from one healthy donor who refrained from eating and drinking two hours before collection, for 1 min at room temperature in polypropylene containers, after obtaining ethical approval from the Research Ethics Committee at the University of Sharjah (#REC-21-10-4-S). Then, samples were randomly allocated to G2–G8 groups according to different decontamination methods as given in Table 1 and Figure 1. 

### 2.3. Fracture Resistance 

The flexural strength and modulus of the samples were tested using the 3-point bending test (*n* = 10 per group). The load was applied at 1 mm/1 min until failure, using the universal testing machine (Tinius Olsen 5ST UTM, Horsham, PA, USA). The dimension of each sample was measured before testing using a caliper. The flexural strength and modulus were calculated using the machine software (Horizon software, Horsham, PA, USA). The broken samples were used for microstructure and surface roughness measurements. 

### 2.4. Microstructure 

One specimen from each group was gold-coated for 120 s using quorum technology Q150TS sputter coater with argon sputtering gas under 10-2 mbar chamber pressure. Then, the microstructure was tested using field emission scanning electron microscope (FE-SEM) Thermoscientific Apreo C (Thermo Fisher Scientific, Waltham, MA, USA).

### 2.5. Surface Roughness (Ra) 

Surface roughness was measured for all tested groups (*n* = 5 per group) using Atomic Force Microscope (AFM, Nanosurf AG, Flex-Axiom with C3000 controller, Gräubernstrasse, Liestal, Switzerland) under a non-contact mode at a scanning speed of 1 s/line and scanning points of 256/line. The scanned area was 10 µm. 

### 2.6. Statistical Analysis 

Using the Statistical Package for Social Sciences Software (SPSS, version 28, Chicago, IL, USA), ANOVA tests were utilized to compare different treatment surfaces while independent *t*-tests were used to compare group means for flexural strength and surface roughness. The *p*-value of <0.05 was considered significant. 

## 3. Results

### 3.1. Fracture Resistance 

Figure 2 shows the means and standard deviations of flexural strength and modulus of different groups of monolithic and multilayered zirconia. Generally, for both types of zirconia, G0 and G1 had similar flexural strength and modulus. Regarding the monolithic zirconia, all tested decontamination methods did not have a significant effect on the flexural strength except phosphoric acid and low-speed bur groups that significantly increased the flexural strength (*p* < 0.05) when compared to the sandblasted/uncontaminated control group—Figure 2a. However, all tested decontamination methods significantly increased the flexural strength of multilayered zirconia (*p* < 0.05) except the phosphoric acid and Al_2_O_3_ sandblasted group that showed no significant difference from the sandblasted/uncontaminated control group—Figure 2c. The flexural modulus of both types of zirconia was not significantly affected by different decontamination methods—Figure 2b,d. The multilayered zirconia showed statistically significant lower flexural strength, but higher modulus than monolithic zirconia. 

### 3.2. Microstructure 

Figure 3 shows SEM images of monolithic and multilayered zirconia, respectively, after decontamination with different methods. Generally, for both types of zirconia, the as-received groups are characterized by the presence of well-defined irregular grains—Figure 3I,II(a), respectively. After sandblasting of monolithic the grains become bigger in size than those in the G0 group—Figure 3I(b). After saliva contamination and water decontamination, the grains appeared irregular and smaller in size—Figure 3I(c). After the use of ZirClean^TM^ or phosphoric acid, the grains become more irregular and smaller than those seen in G0–G2—Figure 3I(d,e), respectively. After decontamination with hydrofluoric acid, the heterogeneity of the surface increased with the presence of both large elongated and very small spherical grains—Figure 3I(f). Decontamination with 50 µm Al_2_O_3_ produced sod-like spherical crystals on the surface—Figure 3I(g). Decontamination of the zirconia surface with the low-speed burs introduced deep surface flaws and defects—Figure 3I(h). With high-speed burs, however, the grains become spherical and smaller in size than those in G1—Figure 3I(i). 

Regarding the multilayered zirconia, after sandblasting the as-received group, the grain showed more irregularities in both size and morphology—Figure 3II(b). After saliva contamination and water decontamination, both large elongated and very small spherical grains were detected Figure 3II(c). After the use of ZirClean^TM^, an increased surface heterogeneity and irregularity in grain size and morphology were observed—Figure 3II(d). After decontamination with phosphoric acid, increased irregularity was observed—Figure 3II(e). After decontamination with hydrofluoric acid, the surface became highly heterogenous and irregular-shaped grains were observed—Figure 3II(f). After decontamination with 50 µm Al_2_O_3_, the surface showed the presence of small spherical grains—Figure 3II(g). Decontamination with low-speed burs introduced deep surface flaws and defects—Figure 3II(h)—while high-speed burs s showed relatively shallow irregularities and introduced crack lines on the surface—Figure 3II(i). 

### 3.3. Surface Roughness (Ra)

AFM images of monolithic and multilayered zirconia subjected to different decontamination methods are shown in Figure 4. Generally, the as-received group showed a smooth surface—Figure 4I,II(a)—while more heterogeneity and irregularity were observed in the remaining groups—Figure 4I,II(b–i). 

Regarding monolithic zirconia, sandblasting the as-received group with 50 µm Al_2_O_3_ significantly increased the surface roughness. Different decontamination methods did not change the surface roughness when compared with G1 control except hydrofluoric acid and low-speed bur that showed a significant increase in surface roughness (*p* < 0.05)—Figure 5a. Regarding the multilayered zirconia, all tested decontamination methods significantly increased the surface roughness (*p* < 0.05), except the high-speed bur group that showed no significant difference from the sandblasted/uncontaminated control. The increase in surface roughness however with G2 group is not expected and warrant further investigation —Figure 5b. 

## 4. Discussion

Zirconia has great affinity to phosphate ions, found in saliva and blood, that encounter the restoration surface during try-in [17]. These phosphate ions bind to the surface of zirconia through ionic bonds and they cannot be removed by water rinsing [17,18]. The presence of adsorbed ions on the surface could explain the significant increase in roughness of multilayered zirconia following decontamination with water compared to the sandblasted/uncontaminated control, but further investigation is still required to confirm this explanation particularly this result was not expected and observed only with multilayered but not monolithic zirconia. Such strong bonds formed on the surface of zirconia restorations could also account for the significant increase in the flexural strength of multilayered zirconia decontaminated with water when compared to the sandblasted/uncontaminated control. However, surface roughness, flexural strength, and flexural modulus of monolithic zirconia were not affected. 

As reported in previous studies, complete decontamination of the zirconia surface cannot be achieved with water rinsing, saliva needs to be attacked chemically by a strong alkaline such as ZirClean^TM^ [19,20]. The active ingredient of ZirClean^TM^ is potassium hydroxide (KOH) that has a pH > 13. These highly alkaline cleaners work by interrupting or neutralizing the existing ionic bonds formed between salivary phosphate ions and the zirconia surface [18]. KOH reacts with the zirconia surface and produces orthocarbonate potassium (K_4_ZrO_4_), and this could explain the significant increase in irregularity and surface roughness, particularly with multilayered zirconia, after decontamination with ZirClean^TM^ when compared with the sandblasted uncontaminated control. 

Decontamination with 37% phosphoric acid increased the surface roughness of both types of zirconia; the significant increase was observed with multilayered zirconia compared to the sandblasted/uncontaminated control. This increase in surface roughness could be correlated to the great chemical affinity of zirconia to phosphate ions found in phosphoric acid as is the case with those in salivary proteins. The formation of chemical bonds on the zirconia surface could be responsible for increased surface heterogeneity and roughness [21]. However, the increase in surface roughness after ZirClean^TM^ or phosphoric acid did not adversely affect the flexural strength and modulus of both types of zirconia. Nevertheless, a significant increase in the flexural strength of multilayered zirconia was observed with ZirClean^TM^, and this could be related to the formation of orthocarbonate potassium (K_4_ZrO_4_), but further investigation is needed to support this finding. On the other hand, with phosphoric acid, the increased flexural strength was observed with monolithic zirconia. 

Decontamination with 9.5% hydrofluoric acid significantly increased the surface roughness of both types of zirconia. Hydrofluoric acid interacts with salivary organic contaminants and dissolves them and produces nanoporosities that increase the surface irregularity and thus the roughness of both types of zirconia when compared to the sandblasted/uncontaminated control [18,22,23]. Regarding the flexural strength, a significant increase was observed with multilayered zirconia. As observed in a previous study, hydrofluoric acid etching increased the monoclinic crystal phase content on the surface of zirconia [24], and this could explain the increase in flexural strength of multilayered zirconia when compared to the sandblasted/uncontaminated control. As observed in a previous study, this monoclinic crystal phase transformation is only superficial [23]. According to the results of the present study, monolithic zirconia is stronger and more resistant to surface changes than multilayered zirconia, therefore the action of hydrofluoric acid was not significant on monolithic zirconia compared to the sandblasted uncontaminated control. 

Decontamination with 50 µm Al_2_O_3_ significantly increased the surface roughness of the multilayered zirconia that contained a higher amount of yttria (9.3 ± 0.3 mol%) to stabilize the cubic crystal content at room temperature. Increased amounts of yttria and cubic crystals were attributed to the good esthetics but low strength of the material. Therefore, sandblasting with Al_2_O_3_ particles produces greater effect on the surface roughness of multilayered compared to monolithic zirconia. Highly translucent zirconia showed more pronounced topographic changes and increased surface irregularity compared to conventional zirconia [25]. Changes in surface roughness did not adversely affect the flexural strength and flexural modulus of both types of zirconia as observed in a previous study [26]. 

The decontamination with low-speed bur significantly increased surface roughness and flexural strength of both types of zirconia. On the contrary, high-speed bur decreased surface roughness of both types of zirconia but increased the flexural strength of multilayered zirconia. The change in roughness could be correlated to the diamond bur grit size, with coarse grinding showing higher surface roughness than fine grinding [27]. In this study, the low-speed bur has a medium grit, and the high-speed bur has fine grit size. This could explain the difference in surface roughness obtained with both burs when compared to the sandblasted uncontaminated control. Decontamination with medium-grit low-speed diamond produced deep surface defects that significantly increased the surface roughness of both types of zirconia. Increased surface roughness did not adversely affect the flexural strength and flexural modulus of both types of zirconia. A significant increase in the flexural strength of monolithic zirconia was observed with low-speed bur while multilayered zirconia showed a significant increase in flexural strength with both low-speed and high-speed burs. The increase in flexural strength could be attributed to the transformation toughening phenomenon induced by the mechanical stress produced during the grinding of zirconia with burs [28]. The mechanical stresses, produced from grinding, induced the transformation of zirconia crystals from tetragonal to monoclinic phase with the accompanying volumetric expansion and an increase in the compressive stresses in the area. These compressive stresses will prevent crack propagation and this phenomenon is known as transformation toughening, which is consequently associated with an increase in flexural strength [28]. 

Due to the importance of surface roughness of the fitting of the zirconia crowns in enhancing the bond strength with the luting cement [29], the effect of different decontamination methods on surface roughness was investigated. Multilayered zirconia showed significantly lower surface roughness and flexural strength than monolithic zirconia. This could be related to the variation in composition of both materials. The increased yttria content, cubic phase crystals content, and grain size in multilayered zirconia are responsible for improving the translucency of the material while mechanical properties are deteriorated, resulting in a weaker material [25,30]. Increased translucency and esthetics in multilayered zirconia are directly associated with the decreased surface roughness of the material as smooth surfaces reduce the loss of incident light. Therefore, generally, the multilayered zirconia has lower surface roughness than monolithic type. Even though the multilayered zirconia showed less roughness than the monolithic type, it showed more pronounced changes in surface roughness than the monolithic type and this could explain the significantly decreased flexural strength of multilayered zirconia compared to monolithic zirconia [31]. The findings in this study are in agreement with previous studies [32,33]. 

Flexural strength and modulus are important properties for the restoration to resist the occlusal load [34]. Higher strength and modulus indicate a more reliable material in clinical conditions [30]. The flexural strength is defined as the maximum amount of applied bending force before failure of the material occurs and the modulus depends on the monoclinic phase content, flaw size, and distribution [35]. High modulus is expected with a specific volume of a material that has low variation in flaw sizes (i.e., defects and flaws are uniform and evenly distributed throughout the structure) [30]. As observed in this study, the flexural modulus of multilayered zirconia was higher than in monolithic zirconia, and it was not caused by different decontamination methods. These findings might indicate that the effects of different decontamination methods occur only on zirconia’s surface.

Based on the findings of this study, it can be recommended that a low-speed bur could be used for decontamination of a contaminated zirconia surface, as it significantly increases the surface roughness and flexural strength. A limitation of this in vitro study is that findings are based on laboratory experimental conditions that may not answer complex clinical questions. 

## 5. Conclusions

Within the limitations of this in vitro study, it was concluded that multilayered zirconia is weaker, and its surface is less resistant to change than monolithic zirconia. In clinical settings following saliva contamination during the try-in stage, decontamination with low-speed bur can be beneficial as it significantly increases the surface roughness as well as the flexural strength of both types of zirconia. The effect of different decontamination methods on the contact angle, shear bond strength, and fatigue resistance will be considered in our future work. For this purpose, zirconia crowns will be used. 

## Figures and Tables

**Figure 1 materials-16-02356-f001:**
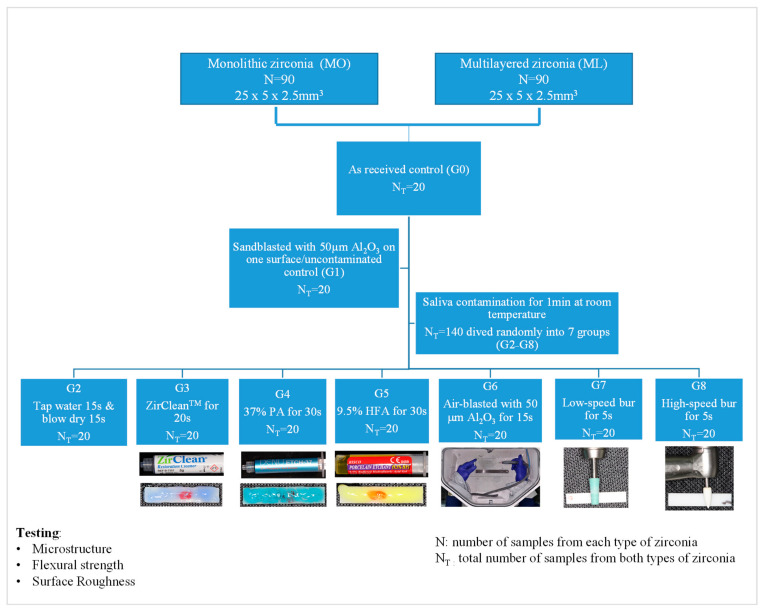
Flow diagram showing different groups used in this study.

**Figure 2 materials-16-02356-f002:**
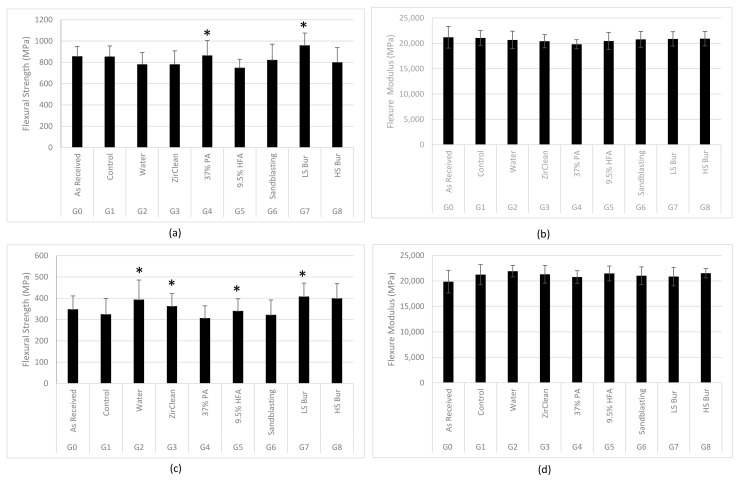
(**a**,**c**) Flexural strength (MPa) of monolithic and multilayered zirconia, respectively. (**b**,**d**) Flexural modulus (MPa) of monolithic and multilayered zirconia, respectively. The *t*-test was used to compare the means of any group and the control. * Represents groups with statistically significant change (*p* < 0.05) in flexural strength compared to control group.

**Figure 3 materials-16-02356-f003:**
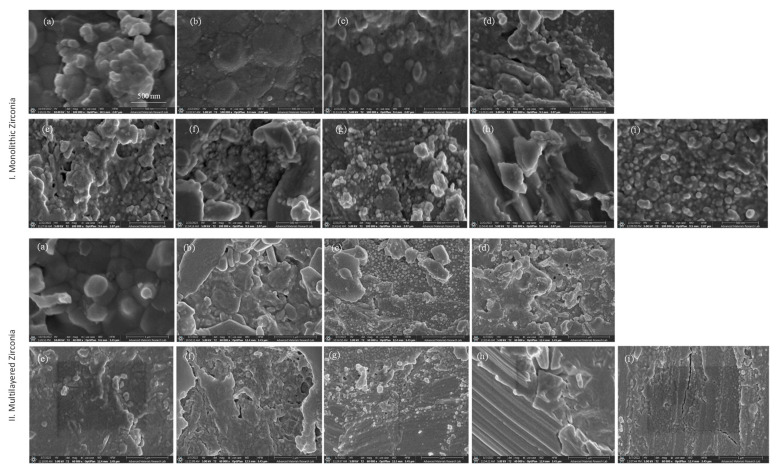
Scanning electron microscopy images of monolithic and multilayered zirconia: (**a**) as received, (**b**) control, (**c**,**d**) ZirClean^TM^, (**e**) 37% PA, (**f**) 9.5% HFA, (**g**) sandblasting, (**h**) LS bur, and (**i**) HS bur. The microstructure was affected according to the decontamination method and the type of zirconia used.

**Figure 4 materials-16-02356-f004:**
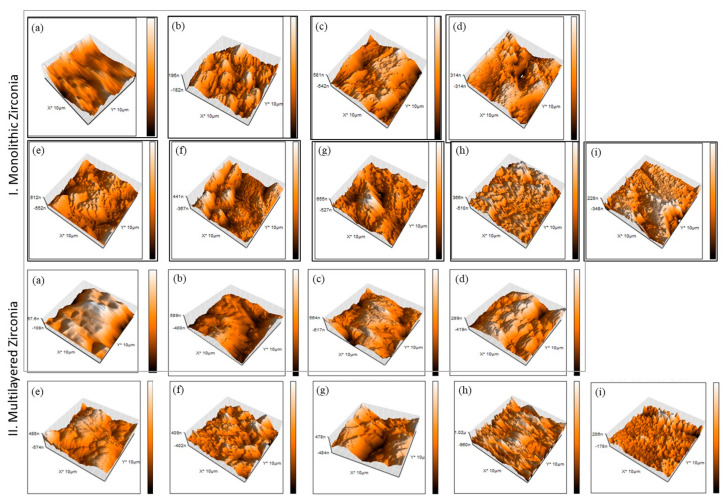
Atomic force microscopy images of monolithic and multilayered zirconia (**a**) as received, (**b**) control, (**c**) water, (**d**) ZirClean^TM^, (**e**) 37% PA, (**f**) 9.5% HFA, (**g**) sandblasting, (**h**) LS bur, and (**i**) HS bur. The surface roughness was affected according to the decontamination method and the type of zirconia used.

**Figure 5 materials-16-02356-f005:**
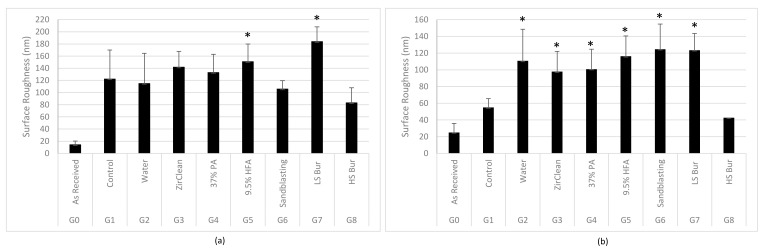
Surface roughness of monolithic (**a**) and multilayered zirconia (**b**). The *t*-test was used to compare the mean of any group and the control. * Represents groups with statistically significant change (*p* < 0.05) in surface roughness compared to control group.

**Table 1 materials-16-02356-t001:** Groups and different treatment methods for decontamination.

Groups (Code)	Decontamination Method
G0 (As received)	As received
G1 (Control)	Sandblasted, uncontaminated
G2 (Water)	As G1, contaminated, and rinsed with tap water for 15 s then blow-dried for 15 s
G3 (ZirClean^TM^)	As G1, contaminated, and cleaned with ZirClean^TM^ for 20 s, rinsed with tap water for 15 s, then blow-dried for 15 s
G4 (37% PA)	As G1, contaminated, and etched with 37% phosphoric acid gel for 30 s, rinsed with tap water for 15 s, then blow-dried for 15 s
G5 (9.5% HFA)	As G1, contaminated, and etched with 9.5% hydrofluoric acid for 30 s, rinsed with tap water for 15 s then blow-dried for 15 s
G6 (Sandblasting)	As G1, contaminated, and air-blasted with 50 μm Al_2_O_3_ at 2.5 bars for 15 s at a distance of 10 cm perpendicular to the surface
G7 (LS bur)	As G1, contaminated, and decontaminated with low-speed dental stone for 5 s
G8 (HS bur)	As G1, contaminated, and decontaminated high-speed dental stone for 5 s

## Data Availability

Data will be available on request.

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
