# Peer review of "Effect of Different Decontamination Methods on Fracture Resistance, Microstructure, and Surface Roughness of Zirconia Restorations—In Vitro Study"

_materials, 2023, doi:10.3390/ma16062356_

Round 1

Reviewer 1 Report

In General, this manuscript investigated an interesting and relevant topic. The paper is good overall; However, some aspects of the introduction and conclusion must be reviewed before publication is possible.

1.         The abstract requires the addition of quantitative results.

2.         Compose a paragraph-length conclusion rather than the present form's point-by-point description.

3.         The conclusion section needs to explain further research.

Author Response

Dear Editor-in-Chief of Materials

The authors would like to thank you very much for providing us a chance to review the manuscript and also would like to thank the reviewers very much for their great effort in reviewing the manuscript and their invaluable comments.

Below are the responses to reviewer comments highlighted in blue under each point. Furthermore, track changes were used to show all the amendment that has been carried out in the manuscript according to reviewers’ recommendations.

Reviewer 1:

Comments and Suggestions for Authors

In General, this manuscript investigated an interesting and relevant topic. The paper is good overall; However, some aspects of the introduction and conclusion must be reviewed before publication is possible.

The authors would like to thank the reviewer for his/her invaluable comments.

The introduction and conclusion have been revised and edited as requested.

Regarding the introduction, the first paragraph was deleted and replaced by the following one for proper connection between paragraphs;

“Due to its superior mechanical properties, zirconia becomes an excellent alternative for metal-ceramic restorations, but due to its high opacity, it was used as a framework that is typically veneered with a glass-ceramic material [1]. It has been shown that this bilayered restorations are highly susceptible to chipping, delamination, or fracture of its veneer [2]. To overcome these drawbacks, monolithic zirconia was introduced [3], and to improve its translucency, changing the yttria content and sintering temperature were attempted [4]. As observed, an increase in yttria content increased the translucency but reduced the strength. Therefore, a new monolithic multilayered zirconia with different color and translucency-gradients has been developed. This type of zirconia consists of outer enamel, two transition layers, and inner body layer. Across these layers, the degree of translucency decreases from the enamel to the body layer, while the color intensity increases [5, 6].”

Furthermore, a subsection of the aim of the study was added as requested by Reviewer 2.

Regarding the conclusion, it was written as one paragraph; future work and recommendations were also added as suggested by other reviewers. The following is the conclusion section:

“Within the limitations of this in-vitro study, it was concluded that multilayered zirconia is weaker and its surface is less resistant to change than monolithic zirconia. In clinical settings following saliva contamination during the try-in stage, decontamination with low-speed bur can be beneficial as it significantly increases the surface roughness as well as flexural strength of both types of zirconia. The effect of different decontamination methods on the contact angle, shear bond strength, and fatigue resistance will be considered in our future work. For this purpose, zirconia crowns will be used.”

  1. The abstract requires the addition of quantitative results.

Done as requested.

  1. Compose a paragraph-length conclusion rather than the present form's point-by-point description.

The conclusion was written as one paragraph; future work and recommendations were also added as suggested by other reviewers. The following is the conclusion section:

“Within the limitations of this in-vitro study, it was concluded that multilayered zirconia is weaker and its surface is less resistant to change than monolithic zirconia. In clinical settings following saliva contamination during the try-in stage, decontamination with low-speed bur can be beneficial as it significantly increases the surface roughness as well as flexural strength of both types of zirconia. The effect of different decontamination methods on the contact angle, shear bond strength, and fatigue resistance will be considered in our future work. For this purpose, zirconia crowns will be used.”

  1. The conclusion section needs to explain further research.

Done as requested. The following statement was added.

“The effect of different decontamination methods on the contact angle, shear bond strength, and fatigue resistance will be considered in our future work. For this purpose, zirconia crowns will be used.”

Reviewer 2:

Comments and Suggestions for Authors

General

  1. The contents of the manuscript were useful and informative.

The authors would like to thank the reviewer for his/her invaluable comments.

  1. Multiple and consistent grammatical, technical and punctuation errors were noted throughout the manuscript.

English editing was done by a native English speaker.

Some of the sentences require rephrasing (to mention a few: line 119, line 139, 367 and so forth) so that it is sounded scientific and improves the quality of the manuscript. The language could be improved, and this manuscript benefited from professional English editing and proofreading. 

Edited as requested. Furthermore, English editing for the whole manuscript was carried out by a native English speaker.

  1. The authors consistently used the words ‘flexure strength’, while they should use ‘flexural strength’ if it is indicated for a noun, its ability to resist deformation under load.

All the words “flexure strength” were replaced by “flexural strength” as requested.

Introduction

  1. The content in the introduction was adequate. But the sentences can be rephrased so that it sounds more scientific. This article required professional English editing and proofreading.

English editing was done as requested.

  1. The was no connection between the first paragraph and the rest of the paragraphs, as it appears not related to the aim of the study. The information regarding elevated temperature degradation is not related as it will not occur intra-orally. The first paragraph also contains multiple spelling mistakes.

The first paragraph was deleted and replaced by the following one as requested;

“Due to its superior mechanical properties, zirconia becomes an excellent alternative for metal-ceramic restorations, but due to its high opacity, it was used as a framework that typically veneered with a glass-ceramic material [1]. It has been shown that this bilayered restorations are highly susceptible to chipping, delamination, or fracture of its veneer [2]. To overcome these drawbacks, monolithic zirconia was introduced [3], and to improve its translucency, changing the yttria content and sintering temperature were attempted [4]. As observed, an increase in yttria content increased the translucency but reduced the strength. Therefore, a new monolithic multilayered zirconia with different color and translucency-gradients has been developed. This type of zirconia consists of outer enamel, two transition layers, and inner body layer. Across these layers, the degree of translucency decreases from the enamel to the body layer, while the color intensity increases [5, 6].”

  1. The last sentence (from lines 64 – 66) does not need to be in the Introduction section.

This last sentence that includes the null hypothesis was removed as requested.

Materials and Methods and Result

  1. ZirClean (lines 13, 61) is a brand name. It should be written with its trademark ZirCleanTM, and whenever possible, the brand name/country should be appropriately cited

All ZirClean words were replaced with “ZirCleanTM” as requested. Furthermore, the brand name/country of productions were done for the rest of different decontamination methods used.

  1. Line 104, ‘Field Emission Scanning 104 Electron Microscope’ does not need to be capitalized

Done as requested.

  1. The mechanical properties of zirconia depend upon characteristics including composition, the ratio of the monoclinic to tetragonal phase, percentage and type of stabilizer. Therefore, the comparison between the two types of zirconia is not valid as both are not the same type – stabilization elements are not the same. 

The authors totally agree with the reviewer, and the difference between the two types of zirconia was highlighted in the manuscript. The comparison between both types was included to explain the differences seen in response of each type to different decontamination methods.

  1. Sample size calculation was not mentioned

The following section was added as requested, and the section number was updated accordingly.

2.1. Sample Size Calculation

Based on previous research [15, 16], sample size was determined to be 10 samples per group (n=10). Since 9 groups from each material were tested, therefore the total number of samples was 180 (90 samples from each material).

  1. The elemental analysis using EDS in this comparison was not meaningful as it does not represent the actual chemical components of both zirconias. The technical data from the supplier should be obtained.

The elemental analysis section was removed from abstract, aim, materials & methods, results and discussion as requested.

  1. It is unclear why the broken samples were used for microstructure, elemental analysis, and surface roughness measurements.

To ensure the consistency of the results by measuring the same samples. At the same time, to reduce the cost.

The surface roughness of the fitting surface of the zirconia crown will enhance the bond strength between the luting cement and the crown (Franz, A., Winkler, O., Lettner, S., Öppinger, S., Hauser, A., Haidar, M., ... & Schedle, A. (2021). Optimizing the fitting-surface preparation of zirconia restorations for bonding to dentin. Dental Materials37(3), 464-476)

 The following statement was added in the discussion section at the beginning of the surface roughness paragraph and the reference was used.

“Due to the importance of surface roughness of the fitting of the zirconia crowns in enhancing the bond strength with the luting cement [29], the effect of different decontamination methods on surface roughness was investigated. Multilayered zirconia showed significantly lower surface roughness and flexural strength”

Ref: Optimizing the fitting-surface preparation of zirconia restorations for bonding to dentin. Dental Materials37(3), 464-476)

Results

  1. Only one sample from each group was used for scanning electron microscopy analysis, therefore, it was unsounded for the author to claim the grain size was irregular, while the grain size of the as-received group was also not regular.

The grain size of as-received group was edited as requested.

  1. The scale for SEM micrographs cannot be seen.

A scale bar was added for SEM images as requested.

  1. Some of the images in Figure 3 are blurred, and the scale bar cannot be read.

The resolution of the image has been improved. Regarding the scale bar, as show in the images, the scanned area was 10 x 10 mm. 

Discussions

-       Adequate

Thank you very much.

Conclusion

-       The conclusion should summarise the result obtained. The authors could write them in proper paragraphs or numbers, rather than in the point form. 

The conclusion was written as one paragraph; the future work and recommendation was also added as suggested by Reviewer 1 and 3 as follows:

“Within the limitations of this in-vitro study, it was concluded that multilayered zirconia is weaker and its surface is less resistant to change than monolithic zirconia. In clinical setting following saliva contamination during the try-in stage, decontamination with low-speed bur can be beneficial as it significantly increases the surface roughness as well as flexural strength of both types of zirconia. The effect of different decontamination methods on the contact angle, shear bond strength, and fatigue resistance will be considered in our future work. For this purpose, zirconia crowns will be used.”

-       The authors may add some sentences of clinical interpretation of the result and recommendations for future studies.

Done as requested; both future work and recommendation was added to the conclusion section as mentioned in the above comment.

References

 Incorrect formatting/ not following the style of the intended journal

Done as requested.

Reviewer 3:

Comments and Suggestions for Authors

The current paper is interesting and important in the Restorative discipline! After proper review, this manuscript has the merit of being published. The authors need to be clear on some points. Please find relevant comments as below:

The authors would like to thank the reviewer for his/her invaluable comments

- The referee suggests writing the word "in vitro" in italics starting from the title and so on throughout the text.

Done as requested.

- In the introduction, keep “phosphoric” instead of “orthophosphoric”, because in the following paragraphs “phosphoric” is mainly used, and it is the right term.

Done as requested.

- Make a new sub-paragraph for the aim, after line 60.

Done as requested.

- In the materials and methods, it is necessary to add a flowchart or diagram of the experiment done, because only the results alone do not help the reader to make a scheme in the mind.

Done as requested.

- Table 1 clearly shows the subdivision of the groups, so do not repeat the full names and abbreviations of the groups again after the first time they have been used; mention only the respective group.

Done as requested.

- In the Statistical Analysis subsection, add p-values for significance.

Done as requested.

- In figure 1, in the respective legend, indicate the statistical test used and the p-value; so, for figure 4 only mention the test statistic used (the p-value here is just mentioned).

Both are done the same way to be consistent and the type of test and p-value were included as requested.

- The discussion section is very long; it is necessary to be revised.

The discussion section was revised and reduced as requested.

- The conclusions section isn’t clear. In my opinion it is better to remove the second and the third points, because is a repetition of the discussion. Also here, the section needs to be revised and simplified mentioning only the key conclusions in a manner to be like a “Keep the message home, at the end of the reading”.

Done as requested; the future work and recommendation were also included as suggested by other reviewers. The following is the conclusion section:

“Within the limitations of this in-vitro study, it was concluded that multilayered zirconia is weaker and its surface is less resistant to change than monolithic zirconia. In clinical setting following saliva contamination during the try-in stage, decontamination with low-speed bur can be beneficial as it significantly increases the surface roughness as well as flexural strength of both types of zirconia. The effect of different decontamination methods on the contact angle, shear bond strength, and fatigue resistance will be considered in our future work. For this purpose, zirconia crowns will be used.”

Reviewer 2 Report

General

1.     The contents of the manuscript were useful and informative.

2.     Multiple and consistent grammatical, technical and punctuation errors were noted throughout the manuscript. Some of the sentences require rephrasing (to mention a few: line 119, line 139, 367 and so forth) so that it is sounded scientific and improves the quality of the manuscript. The language could be improved, and this manuscript benefited from professional English editing and proofreading. 

3.     The authors consistently used the words ‘flexure strength’, while they should use ‘flexural strength’ if it is indicated for a noun, its ability to resist deformation under load.

Introduction

1.     The content in the introduction was adequate. But the sentences can be rephrased so that it sounds more scientific. This article required professional English editing and proofreading.

2.     The was no connection between the first paragraph and the rest of the paragraphs, as it appears not related to the aim of the study. The information regarding elevated temperature degradation is not related as it will not occur intra-orally. The first paragraph also contains multiple spelling mistakes.

3.     The last sentence (from lines 64 – 66) does not need to be in the Introduction section.

Materials and Methods and Result

1.     ZirClean (lines 13, 61) is a brand name. It should be written with its trademark ZirCleanTM, and whenever possible, the brand name/country should be appropriately cited

2.     Line 104, ‘Field Emission Scanning 104 Electron Microscope’ does not need to be capitalized

3.     The mechanical properties of zirconia depend upon characteristics including composition, the ratio of the monoclinic to tetragonal phase, percentage and type of stabilizer. Therefore, the comparison between the two types of zirconia is not valid as both are not the same type – stabilization elements are not the same. 

4.     Sample size calculation was not mentioned

5.     The elemental analysis using EDS in this comparison was not meaningful as it does not represent the actual chemical components of both zirconias. The technical data from the supplier should be obtained.

6.     It is unclear why the broken samples were used for microstructure, elemental analysis, and surface roughness measurements. The surface roughness of the fitting surface of the zirconia crown will enhance the bond strength between the luting cement and the crown (Franz, A., Winkler, O., Lettner, S., Öppinger, S., Hauser, A., Haidar, M., ... & Schedle, A. (2021). Optimizing the fitting-surface preparation of zirconia restorations for bonding to dentin. Dental Materials37(3), 464-476)

Results

1.     Only one sample from each group was used for scanning electron microscopy analysis, therefore, it was unsounded for the author to claim the grain size was irregular, while the grain size of the as-received group was also not regular.

2.     The scale for SEM micrographs cannot be seen.

3.     Some of the images in Figure 3 are blurred, and the scale bar cannot be read.

Discussions

-       Adequate

Conclusion

-       The conclusion should summarise the result obtained. The authors could write them in proper paragraphs or numbers, rather than in the point form. 

-       The authors may add some sentences of clinical interpretation of the result and recommendations for future studies.

References

Incorrect formatting/ not following the style of the intended journal

Author Response

(The authors gave the same response as above.)

Reviewer 3 Report

The current paper is interesting and important in the Restorative discipline! After proper review, this manuscript has the merit of being published. The authors need to be clear on some points. Please find relevant comments as below:

- The referee suggests writing the word "in vitro" in italics starting from the title and so on throughout the text.

- In the introduction, keep “phosphoric” instead of “orthophosphoric”, because in the following paragraphs “phosphoric” is mainly used, and it is the right term.

- Make a new sub-paragraph for the aim, after line 60.

- In the materials and methods, it is necessary to add a flowchart or diagram of the experiment done, because only the results alone do not help the reader to make a scheme in the mind.

- Table 1 clearly shows the subdivision of the groups, so do not repeat the full names and abbreviations of the groups again after the first time they have been used; mention only the respective group.

- In the Statistical Analysis subsection, add p-values for significance.

- In figure 1, in the respective legend, indicate the statistical test used and the p-value; so, for figure 4 only mention the test statistic used (the p-value here is just mentioned).

- The discussion section is very long; it is necessary to be revised.

- The conclusions section isn’t clear. In my opinion it is better to remove the second and the third points, because is a repetition of the discussion. Also here, the section needs to be revised and simplified mentioning only the key conclusions in a manner to be like a “Keep the message home, at the end of the reading”.

Author Response

(The authors gave the same response as above.)

Reviewer 4 Report

1.     In the abstract, the author's inference should be modified appropriately due to there is no experimental analysis of phase content and crystal size from the present work. 

2.     In the abstract, the materials and method should be clarified.

3.     In line 123, it should be low-speed bur (G7) rather than G8.

4.     Please clarify the reasons why yttrium only occur in G0 of multilayered zirconia.  Is the yttrium also completely removed during the contamination or decontamination process?

5.     The reference style should follow the author’s guide.

Author Response

Dear Editor-in-Chief of Materials

The authors would like to thank you very much for providing us a chance to review the manuscript and also would like to thank the reviewers very much for their great effort in reviewing the manuscript and their invaluable comments.

Below are the responses to reviewer comments highlighted in blue under each point. Furthermore, track changes were used to show all the amendment that has been carried out in the manuscript according to reviewers’ recommendations.

Reviewer 1:

Comments and Suggestions for Authors

In General, this manuscript investigated an interesting and relevant topic. The paper is good overall; However, some aspects of the introduction and conclusion must be reviewed before publication is possible.

The authors would like to thank the reviewer for his/her invaluable comments.

The introduction and conclusion have been revised and edited as requested.

Regarding the introduction, the first paragraph was deleted and replaced by the following one for proper connection between paragraphs;

“Due to its superior mechanical properties, zirconia becomes an excellent alternative for metal-ceramic restorations, but due to its high opacity, it was used as a framework that is typically veneered with a glass-ceramic material [1]. It has been shown that this bilayered restorations are highly susceptible to chipping, delamination, or fracture of its veneer [2]. To overcome these drawbacks, monolithic zirconia was introduced [3], and to improve its translucency, changing the yttria content and sintering temperature were attempted [4]. As observed, an increase in yttria content increased the translucency but reduced the strength. Therefore, a new monolithic multilayered zirconia with different color and translucency-gradients has been developed. This type of zirconia consists of outer enamel, two transition layers, and inner body layer. Across these layers, the degree of translucency decreases from the enamel to the body layer, while the color intensity increases [5, 6].”

Furthermore, a subsection of the aim of the study was added as requested by Reviewer 2.

Regarding the conclusion, it was written as one paragraph; future work and recommendations were also added as suggested by other reviewers. The following is the conclusion section:

“Within the limitations of this in-vitro study, it was concluded that multilayered zirconia is weaker and its surface is less resistant to change than monolithic zirconia. In clinical settings following saliva contamination during the try-in stage, decontamination with low-speed bur can be beneficial as it significantly increases the surface roughness as well as flexural strength of both types of zirconia. The effect of different decontamination methods on the contact angle, shear bond strength, and fatigue resistance will be considered in our future work. For this purpose, zirconia crowns will be used.”

  1. The abstract requires the addition of quantitative results.

Done as requested.

  1. Compose a paragraph-length conclusion rather than the present form's point-by-point description.

The conclusion was written as one paragraph; future work and recommendations were also added as suggested by other reviewers. The following is the conclusion section:

“Within the limitations of this in-vitro study, it was concluded that multilayered zirconia is weaker and its surface is less resistant to change than monolithic zirconia. In clinical settings following saliva contamination during the try-in stage, decontamination with low-speed bur can be beneficial as it significantly increases the surface roughness as well as flexural strength of both types of zirconia. The effect of different decontamination methods on the contact angle, shear bond strength, and fatigue resistance will be considered in our future work. For this purpose, zirconia crowns will be used.”

  1. The conclusion section needs to explain further research.

Done as requested. The following statement was added.

“The effect of different decontamination methods on the contact angle, shear bond strength, and fatigue resistance will be considered in our future work. For this purpose, zirconia crowns will be used.”

Reviewer 2:

Comments and Suggestions for Authors

General

  1. The contents of the manuscript were useful and informative.

The authors would like to thank the reviewer for his/her invaluable comments.

  1. Multiple and consistent grammatical, technical and punctuation errors were noted throughout the manuscript.

English editing was done by a native English speaker.

Some of the sentences require rephrasing (to mention a few: line 119, line 139, 367 and so forth) so that it is sounded scientific and improves the quality of the manuscript. The language could be improved, and this manuscript benefited from professional English editing and proofreading. 

Edited as requested. Furthermore, English editing for the whole manuscript was carried out by a native English speaker.

  1. The authors consistently used the words ‘flexure strength’, while they should use ‘flexural strength’ if it is indicated for a noun, its ability to resist deformation under load.

All the words “flexure strength” were replaced by “flexural strength” as requested.

Introduction

  1. The content in the introduction was adequate. But the sentences can be rephrased so that it sounds more scientific. This article required professional English editing and proofreading.

English editing was done as requested.

  1. The was no connection between the first paragraph and the rest of the paragraphs, as it appears not related to the aim of the study. The information regarding elevated temperature degradation is not related as it will not occur intra-orally. The first paragraph also contains multiple spelling mistakes.

The first paragraph was deleted and replaced by the following one as requested;

“Due to its superior mechanical properties, zirconia becomes an excellent alternative for metal-ceramic restorations, but due to its high opacity, it was used as a framework that typically veneered with a glass-ceramic material [1]. It has been shown that this bilayered restorations are highly susceptible to chipping, delamination, or fracture of its veneer [2]. To overcome these drawbacks, monolithic zirconia was introduced [3], and to improve its translucency, changing the yttria content and sintering temperature were attempted [4]. As observed, an increase in yttria content increased the translucency but reduced the strength. Therefore, a new monolithic multilayered zirconia with different color and translucency-gradients has been developed. This type of zirconia consists of outer enamel, two transition layers, and inner body layer. Across these layers, the degree of translucency decreases from the enamel to the body layer, while the color intensity increases [5, 6].”

  1. The last sentence (from lines 64 – 66) does not need to be in the Introduction section.

This last sentence that includes the null hypothesis was removed as requested.

Materials and Methods and Result

  1. ZirClean (lines 13, 61) is a brand name. It should be written with its trademark ZirCleanTM, and whenever possible, the brand name/country should be appropriately cited

All ZirClean words were replaced with “ZirCleanTM” as requested. Furthermore, the brand name/country of productions were done for the rest of different decontamination methods used.

  1. Line 104, ‘Field Emission Scanning 104 Electron Microscope’ does not need to be capitalized

Done as requested.

  1. The mechanical properties of zirconia depend upon characteristics including composition, the ratio of the monoclinic to tetragonal phase, percentage and type of stabilizer. Therefore, the comparison between the two types of zirconia is not valid as both are not the same type – stabilization elements are not the same. 

The authors totally agree with the reviewer, and the difference between the two types of zirconia was highlighted in the manuscript. The comparison between both types was included to explain the differences seen in response of each type to different decontamination methods.

  1. Sample size calculation was not mentioned

The following section was added as requested, and the section number was updated accordingly.

2.1. Sample Size Calculation

Based on previous research [15, 16], sample size was determined to be 10 samples per group (n=10). Since 9 groups from each material were tested, therefore the total number of samples was 180 (90 samples from each material).

  1. The elemental analysis using EDS in this comparison was not meaningful as it does not represent the actual chemical components of both zirconias. The technical data from the supplier should be obtained.

The elemental analysis section was removed from abstract, aim, materials & methods, results and discussion as requested.

  1. It is unclear why the broken samples were used for microstructure, elemental analysis, and surface roughness measurements.

To ensure the consistency of the results by measuring the same samples. At the same time, to reduce the cost.

The surface roughness of the fitting surface of the zirconia crown will enhance the bond strength between the luting cement and the crown (Franz, A., Winkler, O., Lettner, S., Öppinger, S., Hauser, A., Haidar, M., ... & Schedle, A. (2021). Optimizing the fitting-surface preparation of zirconia restorations for bonding to dentin. Dental Materials37(3), 464-476)

 The following statement was added in the discussion section at the beginning of the surface roughness paragraph and the reference was used.

“Due to the importance of surface roughness of the fitting of the zirconia crowns in enhancing the bond strength with the luting cement [29], the effect of different decontamination methods on surface roughness was investigated. Multilayered zirconia showed significantly lower surface roughness and flexural strength”

Ref: Optimizing the fitting-surface preparation of zirconia restorations for bonding to dentin. Dental Materials37(3), 464-476)

Results

  1. Only one sample from each group was used for scanning electron microscopy analysis, therefore, it was unsounded for the author to claim the grain size was irregular, while the grain size of the as-received group was also not regular.

The grain size of as-received group was edited as requested.

  1. The scale for SEM micrographs cannot be seen.

A scale bar was added for SEM images as requested.

  1. Some of the images in Figure 3 are blurred, and the scale bar cannot be read.

The resolution of the image has been improved. Regarding the scale bar, as show in the images, the scanned area was 10 x 10 mm. 

Discussions

-       Adequate

Thank you very much.

Conclusion

-       The conclusion should summarise the result obtained. The authors could write them in proper paragraphs or numbers, rather than in the point form. 

The conclusion was written as one paragraph; the future work and recommendation was also added as suggested by Reviewer 1 and 3 as follows:

“Within the limitations of this in-vitro study, it was concluded that multilayered zirconia is weaker and its surface is less resistant to change than monolithic zirconia. In clinical setting following saliva contamination during the try-in stage, decontamination with low-speed bur can be beneficial as it significantly increases the surface roughness as well as flexural strength of both types of zirconia. The effect of different decontamination methods on the contact angle, shear bond strength, and fatigue resistance will be considered in our future work. For this purpose, zirconia crowns will be used.”

-       The authors may add some sentences of clinical interpretation of the result and recommendations for future studies.

Done as requested; both future work and recommendation was added to the conclusion section as mentioned in the above comment.

References

 Incorrect formatting/ not following the style of the intended journal

Done as requested.

Reviewer 3:

Comments and Suggestions for Authors

The current paper is interesting and important in the Restorative discipline! After proper review, this manuscript has the merit of being published. The authors need to be clear on some points. Please find relevant comments as below:

The authors would like to thank the reviewer for his/her invaluable comments

- The referee suggests writing the word "in vitro" in italics starting from the title and so on throughout the text.

Done as requested.

- In the introduction, keep “phosphoric” instead of “orthophosphoric”, because in the following paragraphs “phosphoric” is mainly used, and it is the right term.

Done as requested.

- Make a new sub-paragraph for the aim, after line 60.

Done as requested.

- In the materials and methods, it is necessary to add a flowchart or diagram of the experiment done, because only the results alone do not help the reader to make a scheme in the mind.

Done as requested.

- Table 1 clearly shows the subdivision of the groups, so do not repeat the full names and abbreviations of the groups again after the first time they have been used; mention only the respective group.

Done as requested.

- In the Statistical Analysis subsection, add p-values for significance.

Done as requested.

- In figure 1, in the respective legend, indicate the statistical test used and the p-value; so, for figure 4 only mention the test statistic used (the p-value here is just mentioned).

Both are done the same way to be consistent and the type of test and p-value were included as requested.

- The discussion section is very long; it is necessary to be revised.

The discussion section was revised and reduced as requested.

- The conclusions section isn’t clear. In my opinion it is better to remove the second and the third points, because is a repetition of the discussion. Also here, the section needs to be revised and simplified mentioning only the key conclusions in a manner to be like a “Keep the message home, at the end of the reading”.

Done as requested; the future work and recommendation were also included as suggested by other reviewers. The following is the conclusion section:

“Within the limitations of this in-vitro study, it was concluded that multilayered zirconia is weaker and its surface is less resistant to change than monolithic zirconia. In clinical setting following saliva contamination during the try-in stage, decontamination with low-speed bur can be beneficial as it significantly increases the surface roughness as well as flexural strength of both types of zirconia. The effect of different decontamination methods on the contact angle, shear bond strength, and fatigue resistance will be considered in our future work. For this purpose, zirconia crowns will be used.”

Reviewer 4:

  1. In the abstract, the author's inference should be modified appropriately due to there is no experimental analysis of phase content and crystal size from the present work. 

The abstract has been modified after removing the information related to the elemental analysis.

  1. In the abstract, the materials and method should be clarified.

Different sections of the abstract have been modified.

  1. In line 123, it should be low-speed bur (G7) rather than G8.

Modified as requested.

  1. Please clarify the reasons why yttrium only occur in G0 of multilayered zirconia.  Is the yttrium also completely removed during the contamination or decontamination process?

The elemental analysis section was removed as recommended by other reviewers accordingly, the information related to Yttrium and other elements were removed.

  1. The reference style should follow the author’s guide.

Done as requested.

Round 2

Reviewer 2 Report

I have no more comments